# A Novel Dataset for Nuclei Segmentation in Melanoma Histopathology

**Mark Schuiveling**                                         M.SCHUIVELING@UMCUTRECHT.NL
**Willeke A.M. Blokx**                                       W.A.M.BLOKX@UMCUTRECHT.NL
**Gerben E. Breimer**                                        G.E.BREIMER-2@UMCUTRECHT.NL
**Karijn P.M. Suijkerbuijk**                                 K.SUIJKERBUIJK@UMCUTRECHT.NL
*University Medical Center Utrecht*

**Daniel Eek**                                               D.B.EEK@STUDENT.TUE.NL
**Mitko Veta**                                               M.VETA@TUE.NL
*TUE Eindhoven*

**Editors:** Accepted for publication at MIDL 2024

## Abstract

The presence of tumor-infiltrating lymphocytes (TILs) in melanoma is associated with decreased recurrence of primary melanoma and increased survival in metastatic melanoma patients treated with immune checkpoint inhibition. Existing nuclei segmentation models performance is low due to the ability of melanocytes to mimic other cell types and due to existing melanoma specific models utilizing older, sub-optimal techniques. In addition, existing models do not provide tissue annotations necessary for determining the localization of TILs whereas this might also hold predictive value. To address this, we created a melanoma specific dataset with nuclei and tissue annotations. In this paper we describe the methodology used to create the dataset. In addition, we provide preliminary baseline benchmarks.

**Keywords:** Nuclei segmentation, histopathology, melanoma, TILs

## 1. Introduction

Melanoma is an aggressive form of skin cancer with increasing incidence (Rahib et al., 2021). Tumor-infiltrating lymphocytes (TILs) have been linked to decreased recurrence of primary melanoma after excision and with an increased response to immune checkpoint inhibition in metastatic melanoma (van Duin et al., 2023; Chatziioannou et al., 2023; Fu et al., 2019). Currently, the NN192 model is the only clinically validated deep learning TILs scoring model. This model segments nuclei through watershed segmentation followed by a fully connected neural network (Chatziioannou et al., 2023). When compared with watershed segmentation, convolutional neural networks based models show an increase in performance (Kowal et al., 2020; Graham et al., 2019). However, these models are trained on publicly non-melanoma specific datasets leading to misclassifications as melanocytes are known to mimick other cell types (Ronen et al., 2019; Magro et al., 2006). Furthermore, current nuclei segmentation models are not able to determine the localization of TILs (intratumoral, peritumoral or stromal) whereas studies in non-small cell lung cancer and breast cancer show that localization of TILs also holds significant predictive value (Choi et al., 2023; Park et al., 2022). Therefore we created a publicly available dataset with nuclei and tissue annotations in melanoma. With this dataset the aim is to develop nuclei and tissue segmentation models. In this paper we describe the methodology for creating the dataset. In addition, we provide preliminary instance segmentation benchmarks.

## 2. Materials and Methods

**Dataset generation** : Region of interest (ROI) were sampled from H&E stained slides of 100 primary melanomas and 100 metastatic melanomas scanned with a Hamamatsu scanner at 40× magnification (0.23 µm per pixel). From each slide a 40× magnified ROI of 1024×1024 pixels was selected for annotation. In addition, a context ROI of 5120×5120 pixels was sampled to provide information about the broader context for the annotation process and, if needed, to be able to generate a larger amount of annotations. Selection was done by a trained medical expert (M.S.) and subsequently verified by an expert dermatopathologist (W.B.). Manual ROI selection ensured diverse tissue and nuclei types.

Nuclei segmentations were generated with HoverNet pretrained on the PanNuke dataset (Gamper et al., 2019; Graham et al., 2019). Manual annotation was performed by M.S. using Qupath with the following cell categories: tumor, stroma, vascular endothelium, histiocyte, melanophage, lymphocyte, plasma cell, neutrophil, apoptotic and epithelium and tissue categories: tumor, stroma, epithelium, endothelium, necrosis, white background (Bankhead et al., 2017). Annotation categories were based on earlier datasets, in addition we chose categories based on possible predictive value. All annotations were checked by an expert dermatopathologist (W.B.). Intra- and inter-observer agreement (by experienced dermatopathologist G.B.) were determined on 10 randomly selected ROIs.

**Benchmark models:** As baseline benchmark, we performed nuclei segmentation for 4 categories; tumor, lymphocyte, stroma and other (consisting out of all other annotations). These categories are used by the clinically validated NN192 model and make a good first test case. The dataset was split with 60% of primary and metastatic slides used for training, and 40% for a validation (20%) and test set (20%).

Mask R-CNN and HoverNet were trained on our data and compared with annotations generated through inference on the test set by HoverNet trained on the PanNuke dataset (which partly consists of annotated melanoma samples) and the NN192 model. For training data, augmentation consisted of Gaussian blur, random flipping and rotation. In Mask R-CNN anchor box size was set to 8, 16, 32, 64 and 128, score threshold to 0.10, region of interest non-maximum supression (NMS) to 0.2 and region proposal network NMS to 0.5. For HoverNet (PanNuke) the non-neoplastic and apoptotic cell categories were merged into other. Center distance with a threshold of 15 pixels ( 3 $\mu m$) followed by greedy (by score and if not available (NN192) by distance) matching was used to determine TP, FP and FN. (Maier-Hein et al., 2024) For each class $F_1$ score was calculated. Finally to compare models micro $F_1$(aggregation of TP, FP and FN over all classes, followed by $F_1$ score calculation) and macro $F_1$ (the average of class $F_1$ scores) were calculated.

## 3. Results

A total of 95 141 nuclei were annotated in 100 primary melanoma ROIs and 100 metastatic melanoma ROIs. Distribution of nuclei and metastatic tumor location is visualized in Figure 1. In metastatic ROIs more tumor cells were present with plasma cells in relative abundance in lung and lymph node metastasis. In primary ROIs histiocytes were relatively common whereas they were rare in metastatic samples. Most metastatic ROIs were from the skin and lymph nodes. An annotated example from the dataset is in Figure 2.

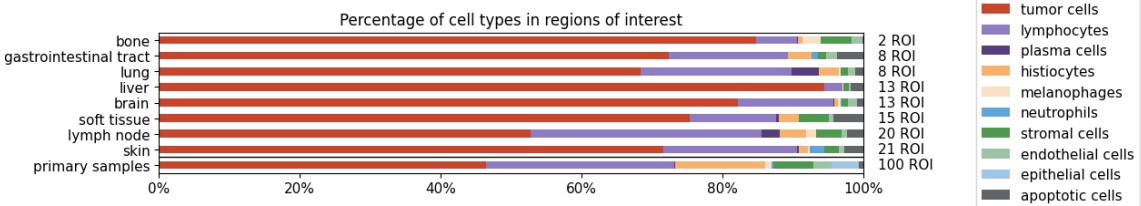

Figure 1: Distribution of cell types in dataset

Table 1: $F_1$ scores for inter/intraobserver and different models

|  | Tumor | Lymphocyte | Stroma | Other | Micro $F_1$ | Macro $F_1$ |
|---|---|---|---|---|---|---|
| **Intraobserver agreement** | 0.97 | 0.90 | 0.75 | 0.84 | 0.94 | 0.86 |
| **Interobserver agreement** | 0.93 | 0.90 | 0.87 | 0.57 | 0.90 | 0.82 |
| **NN192** | 0.61 | 0.12 | 0.06 | 0.03 | 0.47 | 0.20 |
| **Mask R-CNN** | 0.75 | 0.30 | 0.22 | 0.12 | 0.63 | 0.35 |
| **HoverNet (PanNuke)** | 0.72 | 0.37 | 0.20 | 0.14 | 0.57 | 0.36 |
| **HoverNet (Melanoma)** | 0.76 | 0.75 | 0.21 | 0.29 | 0.69 | 0.50 |

Baseline benchmarks are shown in Table 1 and inference results for the dataset example in Figure 2. NN192 and HoverNet (PanNuke) have lower performance due to classifying lymphocytes as tumor cells (in the example) but also due to classifying tumor cells as other and stroma (in the test set). Mask R-CNN has lower performance due to non-maximum suppression hampering detection of tightly packed nuclei. HoverNet (Melanoma) has the highest performance. Intraobserver agreement was high, interobserver agreement was high except for the other category (which consisted out of 421 nuclei) due to the annotation of more tumor cells as apoptotic cells.

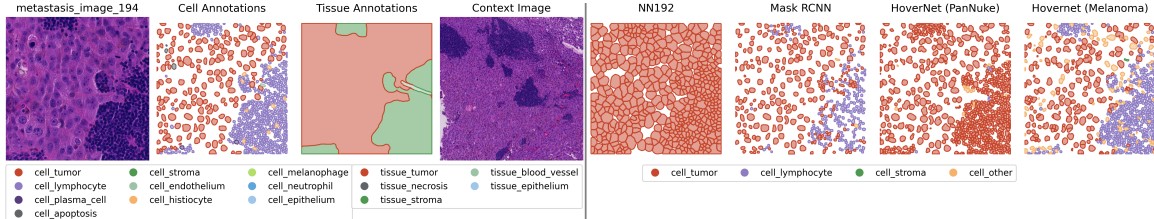

Figure 2: Dataset example of metastasis image 194 with context image and cell and tissue annotations is shown left. Inference results for four categories is shown on the right.

## 4. Conclusion

In this paper we describe the development of a melanoma specific nuclei and tissue segmentation dataset. In addition, we show that training a CNN for nuclei segmentation on a melanoma specific dataset leads to an improvement when compared to HoverNet trained on the PanNuke dataset and the existing melanoma specific NN192 model. Future work will be aimed at evaluating combined nuclei and tissue segmentation to assess to what extent nuclei segmentation of different classes is possible.

The dataset (Schuiveling, 2024) and code (Mask R-CNN, HoverNet) are available online.

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
