# OpenReview forum: "A Novel Dataset for Nuclei Segmentation in Melanoma Histopathology"
_MIDL.io/2024/Short_Papers — MIDL 2024 Short Papers_

### Official Review · Reviewer_TNEE · 2024-04-24

**Confidence:** 5
**Final Rating:** 5

**Review:**

This paper described the development of a melanoma specific nuclei and tissue segmentation dataset. The dataset and code are available online. This is valuable to the research community on developing and evaluating new methods. Different from other cancer types, melanoma presents challenges with more existence of mimics. In summary, this paper is clearly written and the contribution of dataset is significant.

---

### Decision · Program_Chairs · 2024-04-26

Accept